# Superconductivity in Cu Co-Doped Sr_x_Bi_2_Se_3_ Single Crystals

**DOI:** 10.3390/ma12233899

**Published:** 2019-11-26

**Authors:** Aleksandr Yu. Kuntsevich, Victor P. Martovitskii, George V. Rybalchenko, Yuri G. Selivanov, Mikhail I. Bannikov, Oleg A. Sobolevskiy, Evgenii G. Chigevskii

**Affiliations:** 1P.N. Lebedev Physical Institute of the RAS, 119991 Moscow, Russia; victormart57@mail.ru (V.P.M.); rybalchenko.george@gmail.com (G.V.R.); selivan@lebedev.ru (Y.G.S.); darksidds@gmail.com (O.A.S.); e.chigevskii@mail.ru (E.G.C.); 2National Research University Higher School of Economics, 101000 Moscow, Russia; bannikovmi96@gmail.com

**Keywords:** topological insulators, topological superconductors, crystal structure, superconductivity, doping, X-ray diffraction, single crystals

## Abstract

In this study, we grew Cu co-doped single crystals of a topological superconductor candidate Sr${}_{x}$Bi${}_{2}$Se${}_{3}$, and studied their structural and transport properties. We reveal that the addition of even as small an amount of Cu co-dopant as 0.6 atomic %, completely suppresses superconductivity in Sr${}_{x}$Bi${}_{2}$Se${}_{3}$. Critical temperature (∼2.7 K) is rather robust with respect to co-doping. We show that Cu systematically increases the electron density and lattice parameters *a* and *c*. Our results demonstrate that superconductivity in Sr${}_{x}$Bi${}_{2}$Se${}_{3}$-based materials is induced by significantly lower Sr doping level x<0.02 than commonly accepted x∼0.06, and it strongly depends on the specific arrangement of Sr atoms in the host matrix. The critical temperature in superconductive Sr-doped Bi${}_{2}$Se${}_{3}$ is shown to be insensitive to carrier density.

## 1. Introduction

The emergence of unusual superconductivity (SC) in novel topological quantum materials is one of the hottest topics in condensed matter physics and material science. Recently SC was revealed in Weyl semimetals (MoTe${}_{2}$ [1]), topological crystalline insulators (Sn${}_{1-x}$In${}_{x}$Te [2]), atomically thin 2D topological insulators (WTe${}_{2}$,[3],[4]]), Dirac semimetals (2M-WS2 [5], CdAs [6]), and nodal line semimetals (NaAlSi [7]). Discovery of superconductivity in Cu-doped topological insulator Bi${}_{2}$Se${}_{3}$ in 2010 [8] opened a new pathway towards a topological superconductivity (TSC). Among the discussed experimental manifestations of TSC in crystals of this material, there are the zero-bias-feature [9,10,11] (possibly due to Majorana fermion formation) and spontaneous rotation symmetry breaking of the superconducting properties (nematicity) [11,12,13]. Despite progress in crystal growth technology [14,15], Cu${}_{y}$Bi${}_{2}$Se${}_{3}$ suffers from poor crystalline quality, non-uniformity (superconducting volume fraction did not exceed 50%) and degradation in air [8,12].

Its superconductive relative Sr${}_{x}$Bi${}_{2}$Se${}_{3}$, discovered in 2015 [16,17,18], had much better structural quality, demonstrated stability of superconductivity during the storage at ambient conditions, and showed a superconductive volume fraction up to 100% [16,19]. This structural supremacy allowed the discovery of new physics: in particular, resistive nematicity [20,21,22], the Dirac-like character of surface superconducting states [23] and nematic effects above critical temperature [24,25]. Inelastic neutron scattering experiments [26] recently confirmed the *p*-wave low-momentum phonon pairing mechanism in these crystals.

Another feature of Sr${}_{x}$Bi${}_{2}$Se${}_{3}$ is a rather small carrier density ($∼2\times 10^{19}$ cm${}^{-3}$), an order of magnitude less than in Cu${}_{y}$Bi${}_{2}$Se${}_{3}$($∼2\times 10^{20}$ cm${}^{-3}$). Atypically, the elements responsible for superconductivity in both materials electrons host a very small part of the Brillouin Zone. Obviously, small density means the opportunity to control the superconductivity through an external impact.

On a structural level, however, there are many white spots about these materials. First of all, it is still unknown where exactly the metal (Cu and Sr) atoms are located in the lattice and how this placement controls superconducting properties. Why do the elements with so different atomic radii induce SC in Bi${}_{2}$Se${}_{3}$, while the others—do not? This is also puzzling: could it be possible to tune the size of the Fermi Surface and the SC properties by varying the dopant content?

Any design of superconducting material system includes control of its parameters by doping and varying the growth conditions. In well-known cuprates and pnictides, doping changes the volume of Fermi-surface and causes dome-like $T_{c}\left(x\right)$-dependence, where $T_{c}$ is a superconducting transition temperature, and *x* is the dopant content.

For Sr${}_{x}$Bi${}_{2}$Se${}_{3}$ system, no systematic way to move along $T_{c}\left(x\right)$ dependence has been suggested so far. It was shown that average Sr content (x≈0.07) [16,19] in superconductive crystals grown from a melt weakly depends on the nominal Sr content (typically x= 0.1–0.2). However, the fact of superconductivity itself depends strongly on growth conditions [27,28]. Superconductivity and the value of the $T_{c}$ were shown to be very robust: partial iso-structural iso-valent anion replacement of Se by S increases the disorder, broadens the superconducting transition but does not affect $T_{c}$ and carrier density [19].

In our study we, therefore, decided to try a novel strategy: partial cation replacement (Sr by Cu). The logic of such co-doping is the following. For Cu${}_{y}$Bi${}_{2}$Se${}_{3}$ it is known that the electron density (given by the size of the Fermi surface) grows with Cu content [29,30] and saturates at about 2 × 10${}^{20}$ cm${}^{-3}$ for y∼0.1 [14]. So, we expect to tune the Fermi level in Sr${}_{x}$Bi${}_{2}$Se${}_{3}$ by the addition of Cu. This co-doping does not require modification of the sample growth technology. In this paper, we managed to tune the Fermi level by adding Cu; however, this did not lead to a visible effect on $T_{c}$, and eventually, SC was suppressed. Our data, thus, provide better understanding of a Sr${}_{x}$Bi${}_{2}$Se${}_{3}$ system and indicate that $T_{c}$ is Fermi-level-insensitive. We show that the latter fact is in line with the present understanding of the unusual superconductivity in this material.

## 2. Materials and Methods

A series of crystals with various nominal compositions Cu${}_{y}$Sr${}_{x}$Bi${}_{2}$Se${}_{3}$ has been grown using the modified Bridgeman method from the pure elemental components (Bi, Se, Cu (99.999%) and Sr (99.95%)). Elemental components in the desired molar ratios were loaded into the quartz ampoules inside an inert atmosphere glove box, evacuated and sealed. Then, the ampoules were heated at 850 ${}^{\circ }$C for 24 h with periodic stirring followed by a slow cooling to 620 ${}^{\circ }$C at a rate of ∼2 ${}^{\circ }$C per hour. The samples were then annealed at 620 ${}^{\circ }$C for 48 h and water quenched. The crystals obtained by this method had a mirror-like surface and were readily cleaved along the basal plane (see [21] for details).

We obtained SEM-images of the crystal cleaved surfaces and studied the composition using a JSM-7001F scanning electron microscope equipped with an energy-dispersive X-ray analyzer (INCAx-act Oxford Instruments) and INCA software. We used a 25 kV accelerating voltage and beam current of 0.1 mA to collect the EDX spectra. In addition to the peaks from elements targeted, the spectra contained artifacts: Adding up to a few atomic percent in detail—F, Ca, Zn, Ru, Sb, Pt, Hg, Po, Ac, Th, U, etc. Sample composition was calculated, taking into account the relevant matrix elements only.

We used two approaches to elemental analysis: (i) averaging over the sample area, combined with EDX mapping (area about few mm${}^{2}$, typical exposition time 10 h per mm${}^{2}$), and (ii) averaging over several manually selected points within morphologically perfect regions of the crystalline phase (∼10 points per crystal with measurement time about 2000 s per point). The first approach allowed us to compare our results with previous EDX studies [16,19] and make conclusions about the degree of sample non-uniformity (see Results). The second approach allowed us to determine real composition of the crystalline body of the sample.

To characterize the samples obtained, we used single-crystal XRD (Panalytical XPert Pro MRD Extended). Rocking curves on different reflections (predominantly (0 0 6), (0 0 15), (2 0 5)), straightforwardly indicated a degree of in-plane misorientation of the crystal blocks and bending of the crystal surface. Here and below, we use hexagonal notations for Miller indices with an omitted triple index in the basal plane, which is equal to a negative sum of the first two indices.

In order to measure the lattice parameters, we recorded 2θ/ω curves. For precise positioning, the sample was first rotated about the ψ-axis to achieve vertical position for the specified diffraction plane and then shifted along the *x* and *y*-directions of the goniometer to maximize X-ray illumination; hence, the useful signal. We measured the lattice constants with accuracy better than 0.0001 Å, limited by crystal quality. However, as our samples consist of blocks with slightly different lattice parameters, we present the values averaged over several neighboring blocks. *c*-parameter was measured from (0 0 15) reflection peak position, while *a*—from (2 0 5) reflection, using the value of the preliminary determined *c*-parameter.

Electron density and mobility were measured from the Hall effect on cleaved samples of approximately rectangular shape with the dimensions ∼5 × 1 × 0.2 mm. The contacts were glued using silver paint. Transport current density did not exceed 1 mA/mm${}^{2}$, which was shown not to overheat the sample. The AC-measurements (70 to 300 Hz) of the resistivity were performed at temperatures 1.5–300 K in magnetic fields up to ±2 T. We used Quantum Design PPMS-9 and Cryogenics CFMS-16 systems for these measurements. Superconductive transition temperature (if relevant) was determined at the half of the resistivity drop.

## 3. Results

The results of our transport, EDX and structural studies are presented in Table 1. We will discuss below the compositional dependence for each parameter.

### 3.1. SEM and EDX Studies

The effect of doping and co-doping on Bi${}_{2}$Se${}_{3}$ crystal structure is seen already from the SEM images of the cleaved basal plane of the crystals. Parent compound Bi${}_{2}$Se${}_{3}$ (see Figure 1a) has a perfect cleavage plane with a few imperfections. The pristine Bi${}_{2}$Se${}_{3}$ has uniform and almost stoichiometric composition (40.9 atomic % of Bi and 59.1 atomic % of Se), that is nearly independent of the measurement approach (average over the area or by selected points).

On the contrary, doped and co-doped materials are much less uniform. Sr-doped Bi${}_{2}$Se${}_{3}$ is cleaved with many steps; some of them are rather straight and abrupt (Figure 1b). Apparently these steps occur due to internal stress/strain, dislocations or grain boundaries. The formation of such imperfections is caused by the metal excess in the Sr–Bi–Se system with respect to stoichiometric Bi${}_{2}$Se${}_{3}$. Our EDX composition data averaged over a sample area on non-co-doped Sr${}_{x}$Bi${}_{2}$Se${}_{3}$ samples reproduce the results of [16,19] well: Sr content (∼1.3 atomic %; that corresponds to Sr${}_{0.065}$Bi${}_{2}$Se${}_{3}$) is independent of nominal Sr composition (see Averaged composition column for samples 306 and 317 in Table 1).

In Cu co-doped Sr${}_{x}$Bi${}_{2}$Se${}_{3}$ (Figure 1c), although the body of the crystal looks single-crystalline, morphology further degrades: smaller irregular flakes of material are seen on the surface. Presence of non-uniformities as well as huge deviations of the average compositions from the nominal compositions suggest that the crystals contain inclusions of different phases, and averaging the EDX spectra over the sample area is inconsistent. Correspondingly, we performed EDX mapping of our samples.

Figure 2a shows the enlarged SEM image of 328 sample with nominal composition Cu${}_{0.03}$Sr${}_{0.18}$Bi${}_{2}$Se${}_{3}$. Element intensity maps (Figure 2b–e) clearly show that Sr is the most non-uniformly distributed element. The regions with enhanced Sr content (blue circle highlights an example) deviate morphologically. We clearly observed similar mapping in all Sr-doped and Cu-Sr co-doped samples: Sr is contained mostly in structurally deviating parts of the sample; Sr and Bi anti-correlate, whereas Se and Cu are more uniformly distributed.

Apparently, in these samples bulk morphologically perfect parts, rather than small Sr-rich inclusions, are responsible for sharp XRD reflections and transport parameters (see next sections). Shielding volume fraction reaches ∼100% in Sr${}_{x}$Bi${}_{2}$Se${}_{3}$ [16,19], and degrades with co-doping, as we show below. In order to characterise these crystalline parts we performed averaged-by-points EDX measurements. First, for each sample we manually selected ∼10 domains (∼20 μm ×20
μm) with perfect morphology, the absence of extra Sr on the element map and no non-uniformities (see the insert and green crosses in Figure 2a). Then we collected EDX spectrum in the selected points of these domains as described in the Methods section. The EDX composition for Cu, Sr, Bi and Se turned out to be rather reproducible from point to point, as we demonstrate in Table 2. The averaged results of these measurements are summarized in the Crystal composition column of Table 1.

These EDX data allow us to make several conclusions:Cu, Bi and Se content in the crystalline part are much closer to the nominal composition than the values averaged over the cleaved surface. This is probably because the cleave tends to pass through the planes with higher density of inclusions.The average content of Sr is much less than its nominal value, similarly to an observation in [16]. This indicates that the extra Sr either does not participate in solid phase formation or is not even dissolved in the melt.Sr content in the crystalline body was found to be 4–6 times smaller than even the average value. This means that most of Sr is contained in Sr-rich minority inclusions; those are formed from Sr-rich melting, in parallel with Sr-poor crystalline phase. Low Sr concentration in the crystalline bulk might be a clue to the explanation of the small carrier density ($∼2\times 10^{19}$ cm${}^{-3}$) in Sr${}_{x}$Bi${}_{2}$Se${}_{3}$.The Sr content in crystals has a tendency to grow with Cu co-doping level. This is an indicator of interaction between Cu and Sr subsystems, as we consider in the Discussion section.

### 3.2. XRD: Block Structure and Lattice Parameters

XRD 2θ/ω scans Figure 3a–c in pristine, doped and co-doped crystals clearly provide evidence for a tetradymite structure with high crystalline quality. Only maximally co-doped sample 320 with nominal composition Cu${}_{0.05}$Sr${}_{0.19}$Bi${}_{2}$Se${}_{3}$ demonstrates some admixture of the second phase indicated by the arrow in Figure 3c. Increase of disorder with Sr-doping and subsequent Cu co-doping in the Bi${}_{2}$Se${}_{3}$ system is also indicated by relative drop of the amplitudes of (0 0 *l*) reflections with large *l*. The ratio $I_{\left(006\right)}/I_{\left(0027\right)}$ is equal to 14.2 for the Bi${}_{2}$Se${}_{3}$ sample, 20.4 for the $Sr_{0.15}$Bi${}_{2}$Se${}_{3}$ sample and 141.2 for the Cu${}_{0.05}$Sr${}_{0.19}$Bi${}_{2}$Se${}_{3}$ sample.

Block structures of the crystals (except pristine Bi${}_{2}$Se${}_{3}$) might even be seen by eye. To quantify the degree of block misorientation and align the samples for further studies we used rocking curves on various reflections. An example is shown in Figure 3d. It is well seen that typical misorientation of the blocks (horizontal scale) is about 1–2${}^{\circ }$. Difference in positions of peaks from different blocks for (0 0 6) and (0 0 15) reflections indicates that blocks are not only misoriented with respect to each other, but also the relevant lattice parameter (in this case *c*-parameter) slightly varies from block to block. Based on the sample lateral dimensions (about 1–3 mm), we evaluate block dimensions to be in the range (100–500) μm. Previously similar block structure has already been reported for Sr${}_{x}$Bi${}_{2}$Se${}_{3}$ (see, e.g., [21], the supplementary material in [25] and the discussion in [24,31]).

From 2θ/ω scans, the positions of (0 0 *l*) reflections are indistinguishable (compare Figure 3a–c) in all samples. To measure lattice parameters, more precise 2θ/ω scans with third triple–crystal analyzer on selected high quality blocks are needed. Examples of such scans at (0 0 15) reflection are given in Figure 3e. The positions of the reflection maxima allow one to determine the *c*-lattice constant—summarized in Table 1 and plotted in Figure 4a. *c*-lattice parameter (i) expands with Sr-doping to some saturated value (this result was previously obtained also in [16]) and (ii) further increases with Cu co-doping. In plane lattice parameter *a* (summarized in Figure 4b and Table 1) was found from 2θ/ω scans on (2 0 5) reflection.

Thus our high-resolution XRD measurements, along with EDX analysis, clearly show that the structural parameters of the body of the crystals are systematically affected by superimposed Cu co-doping. Both *c* and *a* lattice parameters increase with the co-doping level. The effect of Cu doping on *c* parameter of ternary Sr${}_{x}$Bi${}_{2}$Se${}_{3}$
Δc/(Δy·100%)=0.5 pm/% is comparable of Sr impact on *c*-lattice in binary Bi${}_{2}$Se${}_{3}$
Δc/(Δx·100%)=0.63 pm/% (here and below we use Crystal composition data from Table 1 for these estimates). This is surprising, because ionic radius of Cu ($R_{Cu2+}=$0.73 Å in octahedral arrangement) is much smaller than that for Sr ($R_{Sr2+}=$ 1.18 Å). We believe, therefore, that Cu atoms act indirectly and promote incorporation of Sr in the host lattice from the melt during the crystal growth (see Discussion section).

The effect of Cu doping on the *a* parameter of Sr${}_{x}$Bi${}_{2}$Se${}_{3}$
Δa/(Δy·100%)=0.12 pm/% is about that of Sr in Bi${}_{2}$Se${}_{3}$
Δa/(Δx·100%)=0.1 pm/%. We should note, that *a*-parameter is less reliable because in-plane sample-specific structural distortions are inherent to this material [21,25].

### 3.3. Electrical Transport

Temperature dependencies of the resistivity for representative crystals are shown in Figure 5. All samples demonstrate metallic behavior with RRR below 2, that suggests a rather huge degree of disorder. Non-co-doped Sr${}_{x}$Bi${}_{2}$Se${}_{3}$ and weakly Cu co-doped (nominal y≤0.02) samples are superconductive; however, the superconductivity in Cu co-doped samples might be incomplete (resistivity does not drop to 0, as seen in Figure 5b). This reflects the reduction of superconductive volume fraction with Cu co-doping. Moreover, similarly to Cu${}_{x}$Bi${}_{2}$Se${}_{3}$ [8,12], we observed degradation of Cu co-doped samples with time. Being stored at ambient conditions for several months, the samples lost superconductivity and even changed their surface color from shiny gray metallic to gray-brown, similarly to [8]. SC transition temperature $T_{c}$ does not demonstrate a dependence on doping level (see Discussion section). Cu co-doping also does not affect the nematicity of superconductivity (results on 329 samples are included in Supplementary of [25]).

Well defined sample geometry (see Figure 5c) allowed us to measure carrier density and mobility (from Hall effect) with 15% accuracy. The resultant electron density is shown to grow with Cu content (empty boxes in Figure 6a). This density increase correlates well with the infrared reflectivity data of [29] on Cu${}_{x}$Bi${}_{2}$Se${}_{3}$ samples (stars in Figure 6a). We conclude, therefore, that Cu moves up the Fermi level and provides *n*-type carriers to the system. Electronic structure might be easily triggered by Cu co-doping. Indeed, it is unlikely that Cu substitutes Se atoms in the crystal due to the different chemical nature [29]. Furthermore, STM studies also excluded substitution of Bi by Cu [32]. So, Cu co-dopant either intercalates into the Van der Waals gap, or resides inside quintuple layers, producing interstitial defects. In both cases, Cu atoms act as donors [32]. The values of the mobility (from 100 cm${}^{2}/$Vs to 1000 cm${}^{2}/$Vs) are in good agreement with previous studies on Sr${}_{x}$Bi${}_{2}$Se${}_{3}$ [16,20,21]. The mobility value is not very conclusive because crystals contain inter-block boundaries. Such imperfections may affect, arbitrarily, the transport current redistribution. However, the tendency for the mobility to decrease with *y* is clearly seen from the Table 1.

## 4. Discussion

Let us summarize first what is known about Cu${}_{y}$Bi${}_{2}$Se${}_{3}$ system [14,29,30]. For small *y*, electron density linearly grows with *y* and saturates at a level about $2\times 10^{20}$ cm${}^{-3}$ for nominal composition y∼0.1. Superconductivity emerges for higher values of *y*. Electron density is constant $∼2\times 10^{20}$ cm${}^{-3}$ for all superconductive compositions. For y>0.5, superconductivity weakens and eventually disappears. Formation of a superconductive phase is shown to be related to some internal strain with a threshold-in-*y* character and triggers superconducting pairing mechanisms [33]. Two methods to obtain Cu-intercalated crystals—direct melt growth [8,29] and subsequent electrochemical intercalation [14,15] both produce rather nonuniform samples [11], with different superconductive volume fractions.

Sr${}_{x}$Bi${}_{2}$Se${}_{3}$ is a different system, where already for small x<0.02 some specific arrangement of Sr atoms forms superconducting crystals with higher structural quality compared to SC Cu${}_{y}$Bi${}_{2}$Se${}_{3}$ samples, achieving SC for y>0.1. This small concentration is limited by the solubility of Sr atoms in the crystalline phase. All extra Sr, as seen from our EDX studies, drops into minority phases and plays no role in superconductivity. Fermi level is pinned by Sr in the bulk, and carrier density is rather small $∼2\times 10^{19}$ cm${}^{-3}$.

When Sr and Cu are superimposed in Bi${}_{2}$Se${}_{3}$ matrix, and the amount of Cu is small (*y* below 0.02), these elements seems to act independently: Cu—as an *n*-type dopant, as in Cu${}_{y}$Bi${}_{2}$Se${}_{3}$ [29,34], and Sr—as a reason for superconductivity. Our paper shows that as the Cu content increases above that value, it suppresses superconductivity, probably due to its influence on Sr arrangement.

Based on our observations, along with the previous Hall measurements in Sr-doped Bi${}_{2}$Se${}_{3}$ system, we suggest a "dome-like" phase diagram, shown in Figure 6b, where the horizontal axis is the Hall-effect electron density in Sr${}_{x}$Bi${}_{2}$Se${}_{3}$ system, and the vertical axis is the temperature. In addition to our data, we plotted all available results on $T_{c}\left(n_{Hall}\right)$ in Sr${}_{x}$Bi${}_{2}$Se${}_{3}$ [16,17] and Sr${}_{x}$Bi${}_{2}$Se${}_{3-y}$S${}_{y}$ [19]. Despite intensive studies of this material, rather limited Hall data were published. [27] reported electron density significantly exceeding consolidated results published by all the other groups. Unfortunately, measurement details were not published in the paper. So, we disregard deviating results of [27] here.

Our study demonstrates that we managed to add higher density points to this dome (marked by arrows in Figure 6b). However, an extra copper definitely suppresses superconductivity. This suggests that the role of Cu co-dopants is two-fold: they add up carriers and influence Sr atom arrangement. When the effect on Sr subsystem becomes too strong, superconductivity disappears.

This observation agrees well with recently confirmed low-$k_{z}$-phonon mediated pairing mechanism [26]. Indeed, Sr, presumably hosting the Van der Waals gaps [16,17,28] provides some inter-layer bonds. This coupling, in turn, determines the phonon dispersion, and ultimately, electron-phonon interaction. As Cu intercalates the system, it perturbs the inter-layer coupling.

Our data also suggest that that Cu promotes incorporation of Sr into the Bi${}_{2}$Se${}_{3}$ lattice during the growth from the melt. Indeed, according to Crystal composition EDX data (Table 1), Sr content enhances with addition of Cu (by approximately 25% for Cu${}_{0.05}$Sr${}_{0.19}$Bi${}_{2}$Se${}_{3}$ with respect to non-co-doped sample). Our X-ray data (see Figure 3e) show that Cu content *y* manages *c*-lattice parameter of Bi${}_{2}$Se${}_{3}$ to grow even stronger than Sr composition *x* does alone. The available data on the direct effect of sole Cu on *c*-lattice parameter of Bi${}_{2}$Se3 are rather ambiguous: in [8,15] *c*-parameter grows essentially with addition of Cu; in [35] it almost remains unchanged; in [36] it even decreases. Probably, this ambiguity is related to amphoteric character of Cu-dopant [29] and its small ionic radius. In our case of Sr-doped Bi${}_{2}$Se${}_{3}$, we believe, therefore, that the growth of *c*-lattice parameter, as well as all other *y*-dependent properties at least partially originate from surplus of Sr content in the lattice, provoked by Cu.

We also should discuss here the absence of $T_{c}\left(n\right)$-dependence. According to Bardeen–Cooper–Schreifer (BCS)-theory, $T_{c}$ should grow dramatically with density of states, as it takes place, e.g., in superconductive cuprates. Correspondingly, there are two options. Either the pairing mechanism is unusual (that was suggested in [26]), or density of states weakly depends on the Fermi level. The latter case corresponds to quasi-two-dimensional, cylinder-like Fermi surface. Such Fermi surface topology has already been observed in superconductive Cu${}_{x}$Bi${}_{2}$Se${}_{3}$ [30,34] with larger carrier density. We expect that in Sr${}_{x}$Bi${}_{2}$Se${}_{3}$ quasi-two-dimensional Fermi-surface may also emerge.

We have to discuss here another relative superconducting material Nb${}_{z}$Bi${}_{2}$Se${}_{3}$ [37,38]. SC in this material is air-stable, similarly to Sr${}_{x}$Bi${}_{2}$Se${}_{3}$. Electron density is about $2\times 10^{20}$ cm${}^{-3}$, similarly to Cu${}_{y}$Bi${}_{2}$Se${}_{3}$. The experimental manifestations of this system are somewhat controversial. [38] reports crystals with a very pronounced "nematic" phenomenology, similarly to the best Sr${}_{x}$Bi${}_{2}$Se${}_{3}$ [20,25]. However, this research does not contain any structural (XRD) data; that makes direct comparison with Sr${}_{x}$Bi${}_{2}$Se${}_{3}$ system problematic. [37], on the contrary, reports multi-phase crystals. These crystals demonstrate a growth of *c* lattice parameter with *z* ( dc/dz=2 pm/% for nominal *z* < 0.2) and unchanged *a*-parameter, similar to our observations for Sr${}_{x}$Bi${}_{2}$Se${}_{3}$. The similarities point to the common predominant metal impurity location in the Van der Waals gap as an important ingredient required for superconductivity.

This intercalating dopant natures of Sr, Cu and Nb impurities in the Bi${}_{2}$Se${}_{3}$ matrix seems to be the key factor for superconductivity. Indeed, many other cationic impurities in Bi${}_{2}$Se${}_{3}$, such as Ag [39] or In [40], readily substitute for bismuth and do not bring superconductivity at ambient pressure.

## 5. Conclusions

In our study, we tried Cu co-doping as a novel strategy to tune the properties of topological superconductor candidate Sr${}_{x}$Bi${}_{2}$Se${}_{3}$. Cu co-dopants were shown to serve as additional electron donors. However, increase of carrier concentration did not affect the critical temperature, indicating an unusual character of SC in the system, in-line with numerous experiments by the other groups. We have shown that the crystals are non-uniform and typically contain micrometer-sized, Sr-enriched inclusions. Real Sr concentration within the crystalline body is rather small (∼0.3 atomic %), and as the atomic concentration of Cu co-dopant exceeds this value, superconductivity is suppressed. We relate this effect to structural modification caused by Cu in the Sr–Bi–Se subsystem. 

## Figures and Tables

**Figure 1 materials-12-03899-f001:**
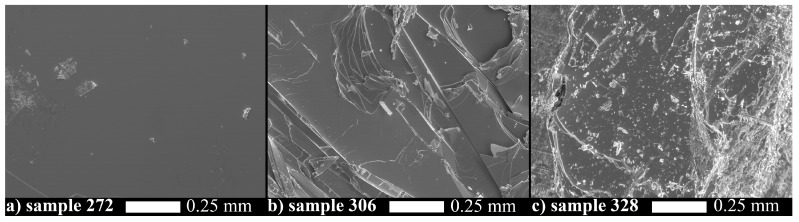
SEM images with 100× magnification of the (001) cleaved surface of the pristine Bi${}_{2}$Se${}_{3}$—sample 272 (**a**), Sr${}_{0.1}$Bi${}_{2}$Se${}_{3}$—sample 306 (**b**), and Cu${}_{0.03}$Sr${}_{0.18}$Bi${}_{2}$Se${}_{3}$—sample 328 (**c**).

**Figure 2 materials-12-03899-f002:**
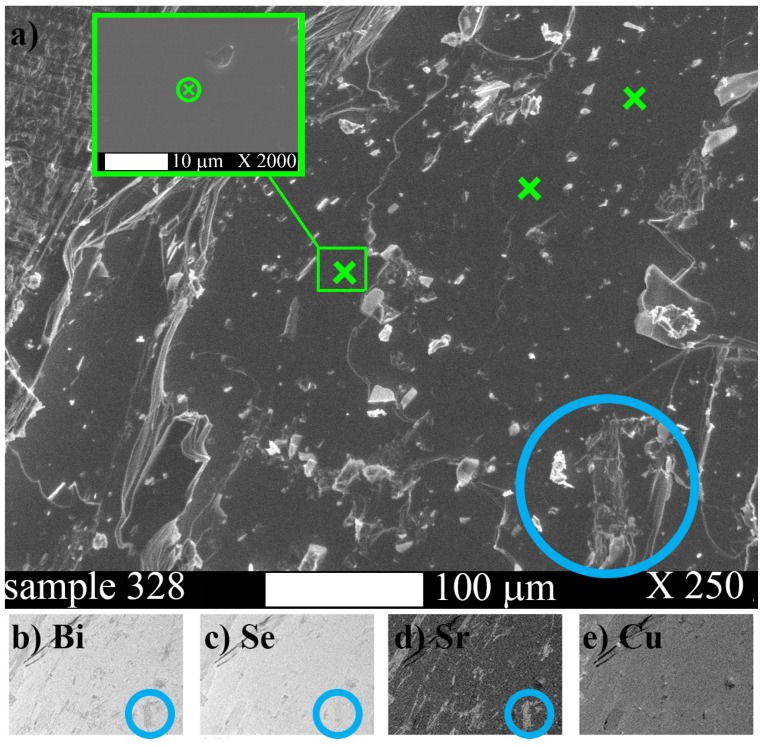
SEM image at 250× magnification (**a**) for sample 328 with nominal composition Cu${}_{0.03}$Sr${}_{0.18}$Bi${}_{2}$Se${}_{3}$. Green crosses show the examples of the points where crystal phase composition was analysed. The insert shows an example of a zoom-in of a relevant region. Panels (**b**–**e**) show the corresponding EDX element distribution maps: (**b**) Bi map; (**c**) Se map; (**d**) Sr map; (**e**) Cu map. Blue circles highlight the representative domains where anti-correlatiosn between Sr and Bi distribution are seen. The maps were collected for 48 h.

**Figure 3 materials-12-03899-f003:**
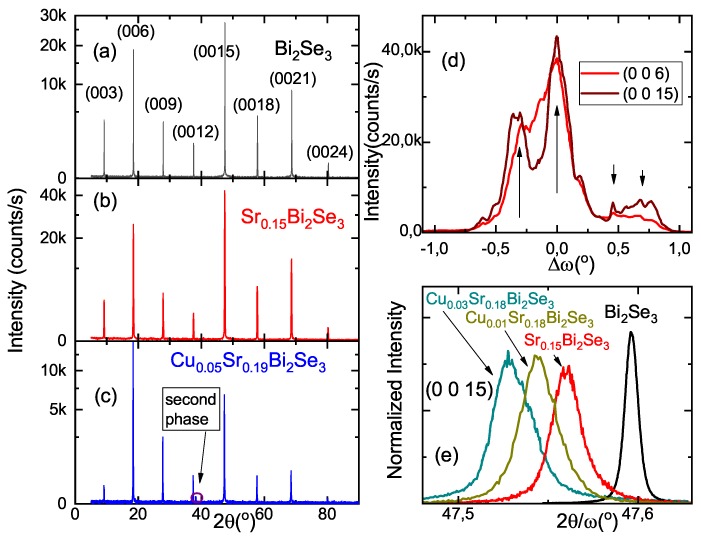
(**a**–**c**) 2θ/ω-scans for three representative samples: (**a**) 272 (Bi${}_{2}$Se${}_{3}$), (**b**) 317 (nominal composition Sr${}_{0.15}$Bi${}_{2}$Se${}_{3}$) and (**c**) 320 (nominal composition Cu${}_{0.05}$Sr${}_{0.19}$Bi${}_{2}$Se${}_{3}$). (**d**) Rocking curve for the multi-block 325 sample (nominal composition Cu${}_{0.01}$Sr${}_{0.18}$Bi${}_{2}$Se${}_{3}$) taken at two reflections—(0 0 6) (red line) and (0 0 15) (wine line). The arrows indicate the angular positions of different blocks. (**e**) 2θ/ω scans at (0 0 15) reflection for different crystals (nominal compositions are indicated in the panel).

**Figure 4 materials-12-03899-f004:**
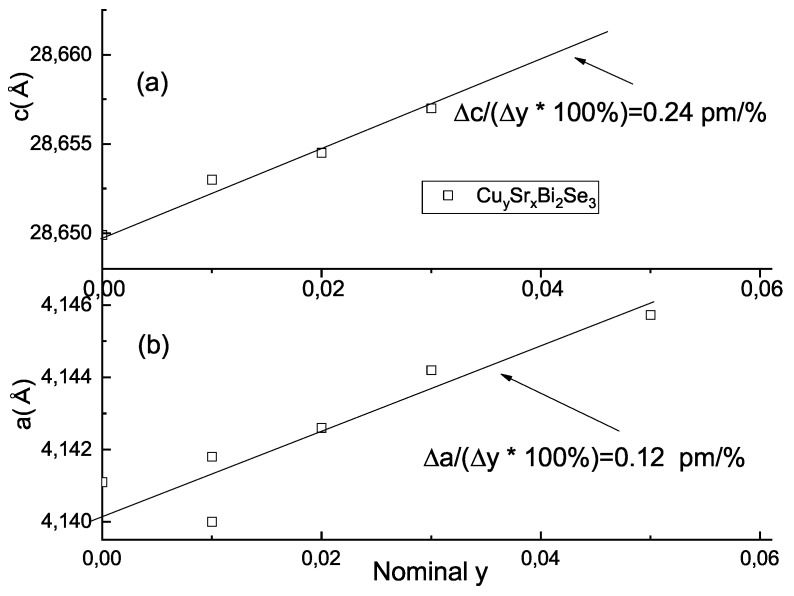
Lattice parameters *c* (panel (**a**)) and *a* (panel (**b**)) as a function of Cu nominal content *y*.

**Figure 5 materials-12-03899-f005:**
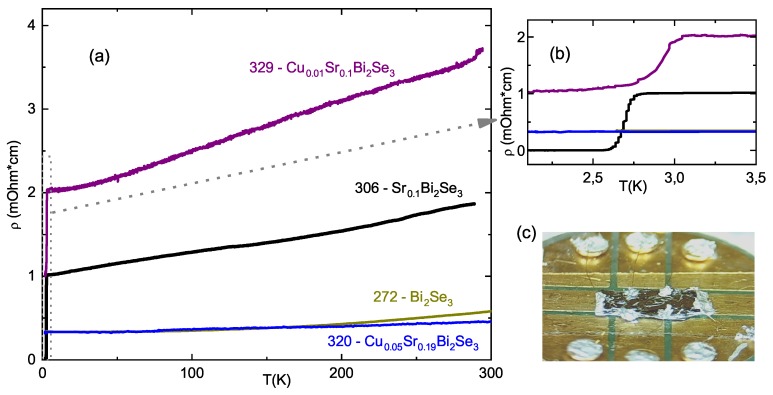
(**a**) Resistivity as a function of temperature for representative samples from our study (nominal compositions are indicated in the panel); (**b**) zoom-in of low-temperature region, where SC transition occurs; (**c**) photo of the sample, mounted for resistivity and Hall-effect measurements.

**Figure 6 materials-12-03899-f006:**
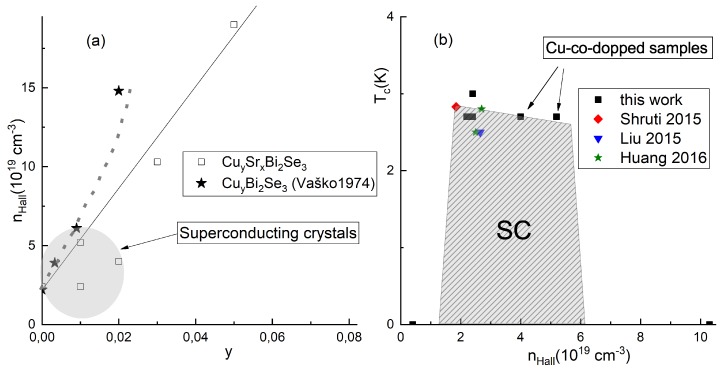
(**a**) Electron density as a function of the nominal co-dopant content *y* in our Cu${}_{y}$Sr${}_{x}$Bi${}_{2}$Se${}_{3}$ samples (empty boxes) compared with Cu${}_{y}$Bi${}_{2}$Se${}_{3}$ reflectivity data of [29] (stars). The straight and dashed lines are just guides to the eye to demonstrate a tendency for the density to grow with *y*. (**b**) Phase diagram of superconductivity ($T_{c}$ versus electron density) according to our results and [16,17,19].

**Table 1 materials-12-03899-t001:** Summary of crystal structure and transport parameters. NSC means that the crystal is not superconductive. Electron density and mobility are collected at T=4 K.

Sample #	Nom. Composition	*n*,10${}^{19}$ cm${}^{-2}$	μ, cm${}^{2}$ /Vs	$T_{c}$, K	RRR	(0015) *c*, Å	(205) *a*, Å	Avg. Composition	Crystal Composition
272	Bi${}_{2}$Se${}_{3}$	0.8	2000	NSC	1.4	28.6343	4.1395	Bi${}_{2.08}$Se${}_{3}$	Bi${}_{2.07}$Se${}_{3}$
306	Sr${}_{0.1}$Bi${}_{2}$Se${}_{3}$	2.2	700	2.7	1.4	28.6596	4.141	Sr${}_{0.064}$Bi${}_{1.58}$Se${}_{3}$	Sr${}_{0.017}$Bi${}_{2.01}$Se${}_{3}$
317	Sr${}_{0.15}$Bi${}_{2}$Se${}_{3}$	2.1	550	2.6	1.4	28.655	4.141	Sr${}_{0.068}$Bi${}_{2.04}$Se${}_{3}$	Sr${}_{0.015}$Bi${}_{1.98}$Se${}_{3}$
329	Cu${}_{0.01}$Sr${}_{0.1}$Bi${}_{2}$Se${}_{3}$	2.4	140	3	1.7	28.6601	4.14	Cu${}_{0.025}$Sr${}_{0.059}$Bi${}_{1.75}$Se${}_{3}$	Cu${}_{0.007}$Sr${}_{0.016}$Bi${}_{2.09}$Se${}_{3}$
325	Cu${}_{0.01}$Sr${}_{0.18}$Bi${}_{2}$Se${}_{3}$	5.2	390	2.7	2	28.6624	4.1418	Cu${}_{0.020}$Sr${}_{0.072}$Bi${}_{1.61}$Se${}_{3}$	Cu${}_{0.017}$Sr${}_{0.016}$Bi${}_{1.62}$Se${}_{3}$
324	Cu${}_{0.02}$Sr${}_{0.15}$Bi${}_{2}$Se${}_{3}$	4.0	390	2.7	1.9	28.6669	4.1426	Cu${}_{0.062}$Sr${}_{0.107}$Bi${}_{1.85}$Se${}_{3}$	Cu${}_{0.028}$Sr${}_{0.018}$Bi${}_{2.07}$Se${}_{3}$
328	Cu${}_{0.03}$Sr${}_{0.18}$Bi${}_{2}$Se${}_{3}$	10.3	167	NSC	1.8	28.6726	4.1442	Cu${}_{0.075}$Sr${}_{0.077}$Bi${}_{1.92}$Se${}_{3}$	Cu${}_{0.039}$Sr${}_{0.016}$Bi${}_{2.07}$Se${}_{3}$
320	Cu${}_{0.05}$Sr${}_{0.19}$Bi${}_{2}$Se${}_{3}$	19	110	NSC	1	28.6617	4.1457	Cu${}_{0.084}$Sr${}_{0.081}$Bi${}_{1.87}$Se${}_{3}$	Cu${}_{0.050}$Sr${}_{0.019}$Bi${}_{2.01}$Se${}_{3}$

**Table 2 materials-12-03899-t002:** Summary on EDX compositional data (in atomic %) taken at six representative points (P1–P6) within the regions with perfect morphology of the (001) surface for the most disordered sample—number 320 (nominal composition Cu${}_{0.05}$Sr${}_{0.19}$Bi${}_{2}$Se${}_{3}$).

Element, Term	P1	P2	P3	P4	P5	P6	Average	Err
Cu K	1.01	0.84	0.73	0.86	1.01	1.47	0.987	0.177
Se K	58.88	59.14	59.22	59.34	59.01	58.88	59.078	0.155
Sr L	0.67	0.37	0.31	0.32	0.27	0.3	0.373	0.099
Bi M	39.44	39.65	39.74	39.48	39.7	39.35	39.560	0.137

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
