# Peer review of "Superconductivity in Cu Co-Doped SrxBi2Se3 Single Crystals"

_materials, 2019, doi:10.3390/ma12233899_

Round 1
Reviewer 1 Report
Kuntsevich et al. describe the doping mechanism of Sr/Cu in the Bi2Se3 system via microscopic analysis of the co-doped samples as well as the pristine samples. The results show that the doping in the Bi2Se3 system increases the inhomogeneity and the Sr content is smaller than the nominal value. Cu co-doping in SrxBi2Se3 increases the carrier numbers, however, the superconductivity eventually suppressed in further doping. Another insight obtained here is that the Tc does not change as the number of carriers increased. This indicates the unique feature of the superconductivity observed in doped Bi2Se3 system.
The analysis is detailed and the comparison between pristine Bi2Se3, SrxBi2Se3, and Cu co-doping is clear. The results inform us of the mechanism of doping in the system and the evolution of structural changes in the doped Bi2Se3 system. The manuscript is reasonably prepared and worth publishing. Below are some points considered before the publication;
There are some studies on the dopant position in Bi2Se3 structure, such as in In-doped samples (K. Kimura et al., Surf. Inter. Anal. 51, 51 (2019).) and Ag-doped samples (T. He et al., Phys. Rev. B 97, 104503 (2018).). Both indicate that the dopant substitutes Bi sites rather than intercalating in the van der Waals gap, thus the situation is different from the current topic. Even so, it is very helpful if the authors can give any insight into the electronic structure triggered by the doping.
For the terminology, the referee is uncomfortable to describe the carrier density obtained from the Hall signal as the Hall density. As all the carriers in the system are electrons, Sr dopant also induces the electrons in doping, it is natural to describe the carrier density as the electron density instead of the Hall density.
The behavior of Tc independent of dopant is also observed in the Nb-doped Bi2Se3. (K. Kobayashi et al., Phys. Rev. B 95, 180503(R) (2017).) The behavior there is deviate from Bi2Se3 structure eventually, however, the behavior is similar, thus it is helpful for the readers to discuss the similarity in the superconductivity in doped Bi2Se3.
Author Response
Please find below our point-by-point response to Reviewer 1 comment (in blue):
There are some studies on the dopant position in Bi2Se3 structure, such as in In-doped samples (K. Kimura et al., Surf. Inter. Anal. 51, 51 (2019).) and Ag-doped samples (T. He et al., Phys. Rev. B 97, 104503 (2018).). Both indicate that the dopant substitutes Bi sites rather than intercalating in the van der Waals gap, thus the situation is different from the current topic. Even so, it is very helpful if the authors can give any insight into the electronic structure triggered by the doping.
Reply: We address this issue in the last paragraph of the"Electrical Transport" section in the revised manuscript, and in the end of the "Discussion" section.
For the terminology, the referee is uncomfortable to describe the carrier density obtained from the Hall signal as the Hall density. As all the carriers in the system are electrons, Sr dopant also induces the electrons in doping, it is natural to describe the carrier density as the electron density instead of the Hall density.
Reply: We replaced "Hall density" by "electron density" throughout the text.
The behavior of Tc independent of dopant is also observed in the Nb-doped Bi2Se3. (K. Kobayashi et al., Phys. Rev. B 95, 180503(R) (2017).) The behavior there is deviate from Bi2Se3 structure eventually, however, the behavior is similar, thus it is helpful for the readers to discuss the similarity in the superconductivity in doped Bi2Se3.
Reply: We discuss the similarities and differences with Nb-doped Bi2Se3 in a new (last but one) paragraph in the “Discussion” section.
Reviewer 2 Report
A.Yu. Kuntsevich report on Cu co-doping of the superconductor SrxBi2Se3. They find that already a small addition of Cu leads to the suppression of superconductivity in this material. They relate this effect to the occurrence of a structural modification, caused by Cu in Sr-Bi-Se subsystem.
The hypothesis of this research is good: Since both CuxBi2Se3 and SrxBi2Se3 are reported superconductors, it is interesting to see what happens if both Sr and Cu are intercalated into the Bi2Se3 structure. Also most of the measurements appear to be well performed and shown (for a few exceptions, see below) in an appropriate manner. The English language of the manuscript is, however, not to the highest standards, and it is often difficult to understand what the scientist want to say. This has to be improved for publication. Beyond that the following science-related questions have to be answered:
The authors write: …”that the superconductivity in SrxBi2Se3-based materials is induced not by doping level, but rather 9 by arrangement of Sr atoms…”. Please add another sentence to the abstract explaining this in greater detail. This statement seems to be rather an assumption than based on experimental factors. The authors must add more evidence to your claims. The introduction is missing a discussion on superconductivity in NbxBi2Se3 and TlxBi2Se3. The manuscript would also benefit from a discussion of other topological non-trivial materials that are superconducting: MoTe2, TaPbS2, Ag2Pb, Sn1−?InxTe, 2M-WS2, NaAlSi, etc. In figure 1, the size bar of the SEM images cannot be read. It must be bigger, while the other information of the machine should be discarded (also true for figure 2). Please box figure 5 (a) for better visualization. Figure 5(c) should be discarded, since it does not contain of any scientific value. Purely Cu doped Bi2Se3, does not become bulk superconducting (Hor PRL), but only electrochemically prepared CuxBi2Se3 shows bulk superconductivity (Ando, PRL). Might the same effect be destroying superconductivity in your samples? Could it be that electrochemical intercalation of Cu into SrxBi2Se3 might actually yield bulk SC samples? Please comment on this. Why is the resistivity of Cu0.01Sr0.1Bi2Se3 higher than of SrxBi2Se3, but the one from Cu0.05Sr0.19Bi2Se3 is lower? Was in figure 5 (a) really sample Cu0.01Sr0.1Bi2Se3 measured, not Cu0.01Sr0.19Bi2Se3? The linear fit to CuySrxBi2Se3 in figure 6 (a) seems most questionable to me. Please explain your train of thought in greater detail. After the brush up of the English language, e.g. through a English language service, I will be better able to comment on the details of the discussions.
Sincerely yours.
Author Response
We reply below the comments by Reviewer 2(shown in blue):
The authors write:
…”that the superconductivity in SrxBi2Se3-based materials is induced not by doping level, but rather by arrangement of Sr atoms…”.
Please add another sentence to the abstract explaining this in greater detail. This statement seems to be rather an assumption than based on experimental factors. The authors must add more evidence to your claims.
Reply: we replaced this misleading statement in the abstract by the one
"Our results demonstrate that superconductivity in SrxBi2Se3-based materials is induced by significantly lower Sr doping level x < 0.02 than commonly accepted x ~ 0.06, and it strongly depends on the specific arrangement of Sr atoms in the host matrix.”
The introduction is missing a discussion on superconductivity in NbxBi2Se3 and TlxBi2Se3.
Reply: Concerning NbxBi2Se3. We are grateful to the Referee for this notion. This material deserves more detailed consideration. We therefore add a paragraph to the Discussion section.
Concerning TlxBi2Se3. We are not aware about superconductivity in this material. Probably, the referee meant TlxBi2Te3. This compound is the only p-type superconducting (at ambient pressure) doped bismuth chalcogenide [Wang et al, Chem. Mater. 2016, 28, 3, 779-784].
There are not so many similarities between TlxBi2Te3 and doped superconducting n-type Bi2Se3. We would prefer to stay within n-type selenide-based systems in our study. Otherwise, our paper transforms into a big review.
The manuscript would also benefit from a discussion of other topological non-trivial materials that are superconducting: MoTe2, TaPbS2, Ag2Pb, Sn1−?InxTe, 2M-WS2, NaAlSi, etc.
Reply: We are thankful to the referee for this note and add two first sentences to the introductory paragraph in order to put our research to broader content, as referee suggests.
In figure 1, the size bar of the SEM images cannot be read. It must be bigger, while the other information of the machine should be discarded (also true for figure 2).
Reply: Done
Please box figure 5 (a) for better visualization.
Reply: Done.
Figure 5(c) should be discarded, since it does not contain of any scientific value.
Reply: We believe, it does. This figure provides the sample geometry and relative sizes of potential contacts. It is this information, that allows us to make the statement on the accuracy:
"...geometry (see Fig. 5c) allowed us to measure carrier density and mobility (from Hall effect) with 15% accuracy"
Purely Cu doped Bi2Se3, does not become bulk superconducting (Hor PRL), but only electrochemically prepared CuxBi2Se3 shows bulk superconductivity (Ando, PRL).
Might the same effect be destroying superconductivity in your samples?
Reply: We are not aware of "the same effect be destroying superconductivity".
In purely Cu doped Bi2Se3 , Cu atoms induce SC, rather than destroy it. Furthermore, starting from x=0.15 SC arises in CuxBi2Se3, and SC is still alive up to x=0.5 [Kreiner2011]. For x<0.1, Cu atoms are unable to induce superconductivity, correspondingly there is nothing to destroy.
Moreover, superconductivity is essentially a bulk property (i.e. not surface) for both melt-grown [Hor2010] and electrochemically intercalated[Kreiner 2011] samples. Both approaches produce samples with a SC volume fraction substantially smaller than 100%.
Probable reason of SC disappearance in our co-doped crystals at y>0.02 is discussed in our manuscript.
Could it be that electrochemical intercalation of Cu into SrxBi2Se3 might actually yield bulk SC samples? Please comment on this.
Reply: We are thankful to Referee for this note and a good idea. Sure, we can't exclude the possibility that electrochemical intercalation of Cu co-dopants could produce SC in CuySrxBi2Se3 with y well above 0.02.
This topic is very interesting, but it is outside the scope of our paper.
The difference between electrochemical intercalation and purely melt growth does not seem to be qualitative. Indeed, melt-growth in Ref. [Hor2010], reports superconducting volume fraction ~20% (Field-cooling to zero field cooling ratio in Fig. 2), whereas chemical intercalation allowed to increase the superconducting volume fraction to 50%. We believe therefore that replacement of direct melt by EC intercalation will not change the fact of presence/absence of superconductivity, but will only affect the SC volume fraction (if it is nonzero).
We add this consideration to Discussion section.
Why is the resistivity of Cu0.01Sr0.1Bi2Se3 higher than of SrxBi2Se3, but the one from Cu0.05Sr0.19Bi2Se3 is lower?
Reply: As we explained in the manuscript, the absolute value of the resistivity and mobility is not very informative, because the crystals consist of rather small blocks. Inter-block boundaries do contribute to resistivity in some uncontrollable manner, as we argued in our previous research [A.Yu. Kuntsevich et al, New. Journ of Phys. 20, 103022 (2018)].
Unfortunately, this effect is unavoidable, especially if large rectangular samples with well defined geometry are cleaved for the Hall effect studies.
Was in figure 5 (a) really sample Cu0.01Sr0.1Bi2Se3 measured, not Cu0.01Sr0.19Bi2Se3?
Reply:Yes. It is easy to see that critical temperature Tc is about 3 K, according to Table 1.
Referee probably expected a smooth and monotonic compositional dependence of the resistivity value, but it is not always the case, as we explain above.
The linear fit to CuySrxBi2Se3 in figure 6 (a) seems most questionable to me. Please explain your train of thought in greater detail.
Reply:This is not a fit. The line is a guide to the eye to show up the tendency for the density to increase with copper content y. This tendency is well comparable with that observed in Ref. [Vasko et al, 1974].
We added a notion to the capture of Fig. 6.
After the brush up of the English language, e.g. through a English language service, I will be better able to comment on the details of the discussions.
Reply: We did our best. And tried to improve the matter!